# Evaluation of Apparent Diffusion Coefficient Repeatability and Reproducibility for Preclinical MRIs Using Standardized Procedures and a Diffusion-Weighted Imaging Phantom

Dariya Malyarenko [1], Ghoncheh Amouzandeh [1,2], Stephen Pickup [3], Rong Zhou [3], Henry Charles Manning [4], Seth T. Gammon [4], Kooresh I. Shoghi [5], James D. Quirk [5], Renuka Sriram [6], Peder Larson [6], Michael T. Lewis [7], Robia G. Pautler [7], Paul E. Kinahan [8], Mark Muzi [8] and Thomas L. Chenevert [1,*]

1   Department of Radiology, University of Michigan, Ann Arbor, MI 48109, USA
2   Neuro42, Inc., San Francisco, CA 94105, USA
3   Department of Radiology, University of Pennsylvania, Philadelphia, PA 19104, USA
4   Department of Cancer Systems Imaging, The University of Texas MDACC, Houston, TX 77030, USA
5   Mallinckrodt Institute of Radiology, Washington University School of Medicine, St. Louis, MO 63110, USA
6   UCSF Department of Radiology & Biomedical Imaging, San Francisco, CA 94158, USA
7   Baylor College of Medicine, Houston, TX 77030, USA
8   Department of Radiology, University of Washington, Seattle, WA 98195, USA
*   Correspondence: tlchenev@med.umich.edu

**Abstract:** Relevant to co-clinical trials, the goal of this work was to assess repeatability, reproducibility, and bias of the apparent diffusion coefficient (ADC) for preclinical MRIs using standardized procedures for comparison to performance of clinical MRIs. A temperature-controlled phantom provided an absolute reference standard to measure spatial uniformity of these performance metrics. Seven institutions participated in the study, wherein diffusion-weighted imaging (DWI) data were acquired over multiple days on 10 preclinical scanners, from 3 vendors, at 6 field strengths. Centralized versus site-based analysis was compared to illustrate incremental variance due to processing workflow. At magnet isocenter, short-term (intra-exam) and long-term (multiday) repeatability were excellent at within-system coefficient of variance, wCV [$\pm$CI] = 0.73% [0.54%, 1.12%] and 1.26% [0.94%, 1.89%], respectively. The cross-system reproducibility coefficient, RDC [$\pm$CI] = 0.188 [0.129, 0.343] $\mu m^2/ms$, corresponded to 17% [12%, 31%] relative to the reference standard. Absolute bias at isocenter was low (within 4%) for 8 of 10 systems, whereas two high-bias (>10%) scanners were primary contributors to the relatively high RDC. Significant additional variance (>2%) due to site-specific analysis was observed for 2 of 10 systems. Base-level technical bias, repeatability, reproducibility, and spatial uniformity patterns were consistent with human MRIs (scaled for bore size). Well-calibrated preclinical MRI systems are capable of highly repeatable and reproducible ADC measurements.

**Keywords:** preclinical MRI; diffusion phantom; repeatability; reproducibility; apparent diffusion coefficient; ADC; ADC bias

## 1. Introduction

Water mobility, quantified via apparent diffusion coefficient (ADC), is being utilized in preclinical and clinical studies as a quantitative MRI biomarker that is sensitive to tissue alteration due to disease evolution and response to treatment [1–4]. ADC measurement has desirable features of being largely independent of magnet field strength, being derived by a simple mathematical model of monoexponential MRI signal decay as a function of diffusion weighting (*b*-value), and being widely available as a standard technique on preclinical and clinical MRI systems. Despite these advantages, disparity in diffusion measurement across sites and scanner platforms has hampered the adoption of ADC as a reliable objective readout of disease/tissue status in (pre-) clinical trials and routine medical practice [5–7]. Aside from biological variability attributable to the subject/patient being scanned, technical

sources undermining ADC reproducibility include variable acquisition protocols, scanner manufacturer and platform capabilities, gradient calibration, and software that convert diffusion-weighted images (DWI) to ADC. Ideally, base-level technical sources of variability are identified, characterized, and mitigated independent of incremental patient-related variability [7–9]. Once an overall level of variability is estimated, realistic confidence thresholds can be established for use of the quantitative biomarker in disease detection, progression, or response to treatment. Degree of variability relative to anticipated effect size has a major impact on study design, feasibility, and financial cost, as well as on scientific expense due to underpowered studies [8–10]. Given this, there is a strong incentive to identify and minimize all technical sources of variability and bias in both clinical and preclinical settings.

Physical phantoms with known properties are essential for technical performance assessments in the quality control (QC) programs [11–14]. Several diffusion phantom materials have been developed over the years, although aqueous solutions of polyvinylpyrrolidone (PVP) are popular and comprise diffusion coefficient standards within homemade and commercially available phantoms [15–18]. PVP is stable and exhibits monoexponential diffusion that is tunable over the full tissue ADC range, although internal phantom temperature must be known and controlled to ≈0.5 °C to measure diffusion coefficients to within 1% accuracy [15,19]. Ice-water-based diffusion phantoms provide an effective inexpensive means for absolute temperature control and a precisely known true diffusion value for MRI system bias assessment [20–22]. Ice-water DWI phantoms have been employed in multicenter clinical studies [22–24] and demonstrate generally good repeatability/reproducibility, reasonable platform and field strength independence, and low absolute bias (≈3%) at magnet isocenter on human scanners [22]. Gradient nonlinearity was identified as the main source of inter-scanner variability and spatial bias patterns as a function of location from isocenter [22,25,26]. Overall good repeatability/reproducibility was also noted previously on preclinical systems, though significant positive bias relative to ground truth was reported [27]. Despite the phantom materials being characterized by specific diffusion coefficients, as opposed to apparent diffusion, the nomenclature "ADC" will be used in this article for consistency with most prior publications.

A central goal of the NCI Co-Clinical Imaging Research Resource Program (CIRP) [28] is to develop quantitative imaging biomarkers applicable to both human and corollary preclinical domains to advance state-of-the-art translational quantitative imaging methodologies from mouse to human. Given its independence of field strength, water diffusion in reference standards should be equivalent on human and mouse MRI systems. The goals of this work were to measure on CIRP preclinical MRI scanners the (1) ADC bias at isocenter; (2) short- and long-term repeatability and cross-system reproducibility; (3) ADC spatial uniformity; and (4) degree of agreement between site-generated ADC versus central-lab-generated ADC values. To achieve these goals, the CIRP image acquisition data processing (IADP) working group (WG) performed a round-robin study of an ice-water-based DWI phantom using a detailed phantom preparation procedure and standardized DWI acquisition protocol, with both site- and core-lab-generated ADC measurements being derived from common DWI datasets.

## 2. Materials and Methods

### 2.1. DWI Phantom

The phantom shown schematically in Figure 1a was constructed from a 50 mL plastic centrifuge tube with a 29 mm outer diameter (OD) lined with a 3 mm thick closed-cell insulation foam and a 100 mm long (8 mm OD) glass measurement tube centrally held in place by foam end plugs. As detailed in the phantom preparation instructions [29], the distilled-water-filled measurement tube was replaced by an air-filled 8 mm OD glass tube, while the phantom interstitial space was filled with water and then frozen overnight in a conventional freezer (−18 °C). The foam insulation lining and end plugs allowed the ice to expand without cracking the plastic centrifuge tube and served to extend the

ice hold time. Immediately prior to scanning, the air-filled glass tube was flushed with 50–60 mL of room-temperature water to melt a thin layer of water so that the air-filled tube could be removed and quickly replaced with the water-filled measurement tube. For RF coils that could accommodate a 45 mm diameter object, the phantom was scanned within an outer foam sleeve (provided with the phantom kit) to further extend the phantom thermal hold time; otherwise, the 29 mm diameter phantom was scanned without the outer foam insulation. Benchtop measurements of temperature versus time following insertion of the measurement tube (initially at room temperature) into the frozen phantom were performed using a 1.37 mm OD optical temperature probe (OTP-M, OPSens, Quebec QC, Canada) located in the center of the measurement tube. Plot of temperature versus time in Figure 1c indicates that the water in the measurement tube quickly achieves thermal equilibrium (<0.5 °C in ≈5 mins) and holds this temperature for at least 90 min, which is sufficient to position the phantom and acquire two sequential DWI scans using the standardized protocol.

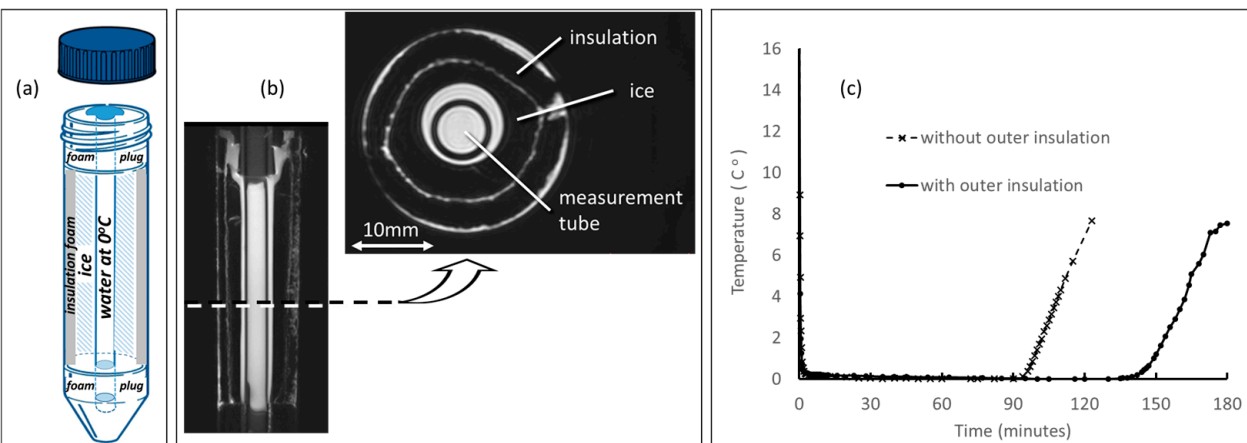

**Figure 1.** (**a**) Schematic of DWI phantom constructed from a 50 mL centrifuge tube designed to hold water in an 8 mm measurement tube at known temperature, thus with known diffusion coefficient. (**b**) Coronal and axial MRI with internal phantom components labeled. (**c**) Plot of measurement tube temperature versus time after measurement tube, initially at room temperature, is inserted into frozen phantom. Note, thermal equilibrium at ≈0 °C is achieved.

### 2.2. DWI Acquisition Protocol

To eliminate potential variability due to acquisition protocol, the CIRP IADP WG achieved consensus on a standardized DWI test procedure that was within the capabilities of all preclinical MRI systems at participating sites. Details of the DWI scan procedure are provided elsewhere [29], though key parameters include Stejskal–Tanner [30] spin-echo DWI sequence ($\Delta = 10$ ms; $\delta = 5$ ms); field-of-view, FOV = 32 mm × 32 mm; acquisition matrix = 64 × 64; 29 axial slices (2 mm thick, 0 mm gap); three-orthogonal DWI directions, target $b$-values = 0, 1000, 2000 s/mm$^2$; number of averages, NSA = 1; and repetition time/echo time, TR/TE = 2000/30 ms for nominal scan duration of 15 min.

### 2.3. Participating Site Procedures

Participating sites were asked to (1) prebuild the standardized DWI acquisition protocol on their MRI system(s); (2) prepare the phantom and scan it twice within a given scan session for intra-exam (short-term) repeatability; (3) repeat the prep/scan process on a second day for inter-exam (long-term) repeatability; (4) provide reconstructed DWI and ADC data in MRI vendor-native format and an insight tool kit (ITK)-compatible format (e.g., DICOM, NIFTI, or MHD) [31–34]; and (5) use its own preferred workflow to generate ADC maps and perform ROI measurements using a 4 mm diameter circular ROI defined within the measurement tube on each slice. This allowed sites to either use scanner-vendor-generated ADC maps, or their own in-house software for off-scanner conversion of DWI

into ADC maps, although site-specific workflow details were not the focus of this study. DWI and ADC maps in vendor-native and ITK-compatible formats from each site were uploaded to the core lab site via shared network storage account (DropBox, per institutional policy), along with each site's ROI measurements. Seven CIRP institutions participated in the study. DWI data were acquired on 10 preclinical MRI systems, from 3 vendor platforms at 6 field strengths. A summary of MRI system demographics and data provided for each system are shown in Table 1. Systems 8 and 9 did not provide the second scans on both days and were excluded from the short-term repeatability evaluation. Inspection of DWI received from all sites (data not shown) confirmed that an adequate cylinder of ice surrounded the measurement tube (Figure 1b), indicating the water was at $\approx 0$ °C, so absolute bias was measurable on all systems.

**Table 1.** MRI system demographics and data produced.

| System | Vendor | Field Strength (T) | Gradient Inner Diameter (mm) | SW Version | Day 1 Scan1 | Scan2 | Day 2 Scan1 | Scan2 | ITK Format |
|--------|--------|--------------------|------------------------------|------------|------|------|------|------|------------|
| 1 | Bruker | 7 | 114 | PV7.0.0 | ✓ | ✓ | ✓ | ✓ | MHD |
| 2 | Bruker | 9.4 | 120 | PV6.0.1 | ✓ | ✓ | ✓ | ✓ | MHD and Classic DICOM |
| 3 | Bruker | 7 | 120 | PV6.0.1 | ✓ | ✓ | ✓ | ✓ | Classic DICOM |
| 4 | Bruker | 9.4 | 114 | PV360 v2.0 | ✓ | ✓ | ✓ | ✓ | Enhanced DICOM |
| 5 | Agilent | 11.74 | 80 | VnmrJ4.2revA | ✓ | ✓ | ✓ | ✓ | Classic DICOM |
| 6 | Bruker | 3 | 105 | PV6.0.1 | ✓ | ✓ | ✓ | ✓ | Classic DICOM |
| 7 | Bruker | 9.4 | 60 | PV360 v3.0 | ✓ | ✓ | ✓ | ✓ | NIFTI |
| 8 | Bruker | 4.7 | 90 | PV6.0.1 | ✓ | | ✓ | | Classic DICOM |
| 9 | Bruker | 14 | 40 | PV5.1 | ✓ | | ✓ | | Classic DICOM |
| 10 | MR Solutions | 3 | 95 | V4.0.2.4 | ✓ | ✓ | ✓ | ✓ | MHD and Classic DICOM |

*2.4. Core Lab Processing*

To mitigate variability in data processing workflow, core lab Matlab version R2019b (Mathworks Inc., Natick, MA, USA) scripts were adapted to convert all sites' vendor-native DWI into ADC maps using a pixelwise linear fit of log DWI signal intensity versus *b*-value, where slope (ADC) and intercept were the fit parameters. Each of three orthogonal DWI directions were fit independently using vendor-provided *b*-values (when available), then averaged for the mean diffusivity (i.e., ADC). Trace DWI (*b* = 0 and geometric mean of 3-orthogonal *b* > 0 DWI) and ADC maps were output in the MHD format. While data input and sort elements of the core lab scripts were tailored for each site datasets, ADC fit routine was held essentially constant. 3D Slicer (version 4.6.2) [35] was used to inspect DWI/ADC MHDs for definition of a 4 mm circular ROI within the measurement tube on each slice independently, then export ROI statistics of the ADC and trace DWI as a function of location along the MRI system *z*-axis. Additional Matlab scripts were used to convert each site's ITK-compatible DWI into ADC, as well as for conversion of the site-generated ADC maps for output as MHDs. Analysis of the core-lab-generated ADC derived from a vendor-native-format DWI was used to measure baseline repeatability/reproducibility for the studied systems, whereas the ADC derived from an ITK-compatible DWI aided interpretation of the potential differences between site-generated and core-lab-generated ADC maps. Low signal-to-noise (SNR) can bias ADC calculation [36,37]; therefore, noise was estimated by the standard deviation (SD) of an ROI drawn in a signal-free background on the first slice, scaled by 1.53 since noise is Rician on magnitude DWI [38]. DWI SNR was

estimated by the mean ROI DWI signal in the measurement tube divided by the noise SD, plotted as a function of location along the *z*-axis (Supplemental Figure S1).

Site-generated ROI mean ADCs (as a function of z-location) were averaged over all available scans (intra- and inter-day exams) from each site's MRI system(s). Likewise, core-lab ROI mean ADC was calculated from the average of core-lab results derived from the corresponding vendor-native-format DWI. Core-lab versus site processing workflows were compared graphically by plotting the relative difference: 100% (ADCsite–ADCcorelab)/Dtrue as a function of *z*-axis location for each system, where Dtrue = 1.1 μm$^2$/ms is the known diffusion coefficient of water at 0 °C [39].

*2.5. Statistics*

The difference in ROI mean ADC from a pair of consecutive DWI scans within each scan session was used to calculate the short-term repeatability. Each site was also instructed to repeat phantom preparation and paired DWI scan acquisition on a second day, yielding two short-term ADC differences (except Systems 8 and 9, Table 1) and two day-to-day differences used to estimate long-term repeatability. For a given pair of ROI mean ADC values ($ADC_1$, $ADC_2$) from the *ith* scanner, mean ($M_i$) and variance ($V_i$) were constructed as [10,40]

$$M_i = \frac{(ADC_1 + ADC_2)}{2}; \quad V_i = \frac{(ADC_1 - ADC_2)^2}{2} \tag{1}$$

Repeatability representative of *N* MRI systems was quantified by estimates of within-system standard deviation (*wSD*), coefficient of variation (*wCV*), and repeatability coefficient (*RC*), defined as [10,40]

$$wSD = \sqrt{\frac{1}{N}\sum_i^N V_i}; \quad wCV = 100\% \cdot \sqrt{\frac{1}{N}\sum_i^N \frac{V_i}{M_i^2}}; \quad RC = 2.77 \cdot wSD . \tag{2}$$

For cross-system reproducibility, available ROI mean ADC values for each system were first averaged across all scans and days for the given system, then mean and standard deviation (*SD*) across the *N* systems was calculated. Analogous to repeatability coefficient, reproducibility coefficient (*RDC*) was assessed as 2.77 *SD*. All repeatability and reproducibility metrics were derived as a function of location along the *z*-axis relative to the magnet isocenter, defined as z = 0. Graphical display of the percent ADC bias was plotted relative to the known diffusion coefficient of water at 0 °C, Dtrue = 1.1 μm$^2$/ms [39], as 100% [(ADC–Dtrue)/Dtrue]. Likewise, *wSD* and *SD* were scaled by 100%/Dtrue on plots so that the degree of variability could be directly compared relative to the systematic bias. Unrealistic ADC values < 0.5 μm$^2$/ms were automatically dropped from the plots and analysis. Each system's absolute ADC bias was measured at the isocenter by averaging the ADC in the measurement tube over the three central slices.

**3. Results**

Measured SNR at the isocenter for low b-value exceeded 150 on all systems evaluated in our study (Supplemental Figure S1a,b). Numerical simulation of noise-induced error using the standardized protocol indicates that bias (i.e., ADC underestimation) would occur for low b-value SNR below 20 (Supplemental Figure S1c).

Figure 2 illustrates the median and range of the absolute ADC bias measured at the isocenter over all scans for each system, with respect to the true diffusion coefficient of water at 0 °C (Dtrue = 1.1 μm$^2$/ms). The small error-bar range relative to the offset from the truth indicates that the system absolute bias was measurable and repeatable on each system. Eight of ten MRI systems were within ±2.5% bias (mostly positive) and the remaining two exceeded +10% bias.

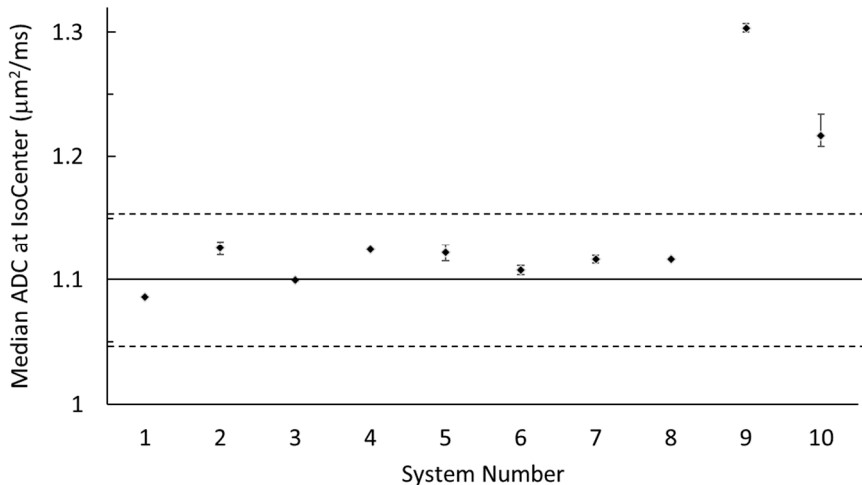

**Figure 2.** Median ADC measured at the isocenter of each system. Each data point is the median of all scans (up to 4) from each system, and error bars indicate the maximum and minimum (range of) isocenter ADC values. The solid line marks Dtrue (1.1 $\mu m^2/ms$), and dashed lines are ±5% relative to Dtrue.

Figure 3 displays relative bias of each system as a function of location on the *z*-axis. All systems are displayed on the same scale to aid visual comparison. Again, the solid horizontal line represents 0% bias relative to Dtrue. Automatic rejection of unrealistic ADC values (<0.5 $\mu m^2/ms$) only occurred outside |z-offset| > 20 mm. Most systems display the pattern of maximum ADC at isocenter, with lower ADC as |z-offset| distance increases, which is consistent with gradient nonlinearity patterns for horizontal bore gradients on human scanners. Systems 9 and 10 showed >10% bias for ADC at isocenter, while others had low isocenter bias (comparable to measurement error). These two systems also showed the greatest gradient nonlinearity over the central region, within ±15 mm of isocenter.

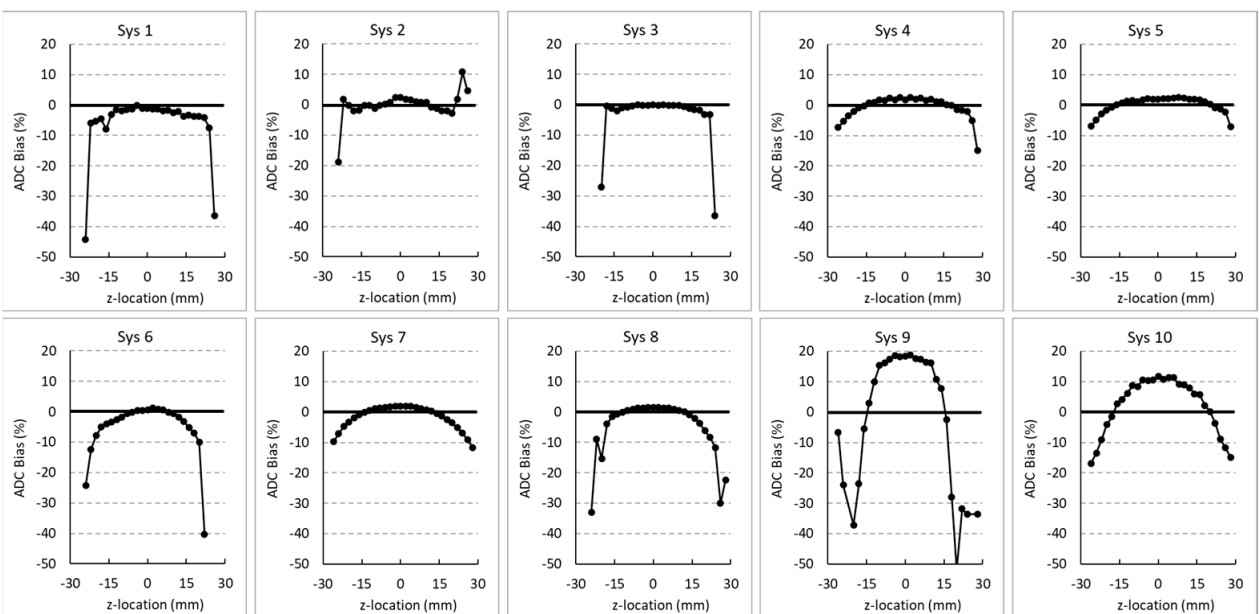

**Figure 3.** Percent bias of each MRI system relative to Dtrue as a function of *z*-axis location.

Plots of the relative bias (with respect to Dtrue), repeatability, and reproducibility of all 10 systems combined are illustrated in Figure 4. Note, relative bias (solid blue line) was unchanged in Figure 4a–c to display the degree of variability (width of shaded region denotes ±100·wSD/Dtrue) relative to bias for short-term (Figure 4a) and long-term

repeatability (Figure 4b) and cross-system reproducibility (Figure 4c). Aggregate bias of ten CIRP systems at the isocenter was within 5% of truth (marked by dashed lines). Short-term repeatability (Figure 4a) wCV relative to Dtrue was <1% for all 10 systems at the isocenter and was fairly uniform with respect to location on *z*-axis. There was a slight increase in wSD observed at the isocenter for long-term repeatability (Figure 4b) that further increases with distance from the isocenter. As expected, reproducibility across all systems (Figure 4c) shows the greatest variance (shaded region denotes $\pm 100 \cdot SD/Dtrue$), with standard deviation (SD) < 7% across systems for locations within $\pm 15$ mm of the isocenter, which increased to 10–15% for greater |z-offset| locations. Summary statistics (with 95% confidence intervals) relevant to ADC measurements at isocenter on all systems are provided in Table 2. The Bland–Altman analysis for isocenter ADC measurements excluding the outliers Sys9 and Sys10 is summarized in Supplemental Figure S2. Without the two outlier systems, the short-term and long-term repeatability were comparable, and cross-system reproducibility was within 3%, with 1% average bias (Supplemental Figure S2).

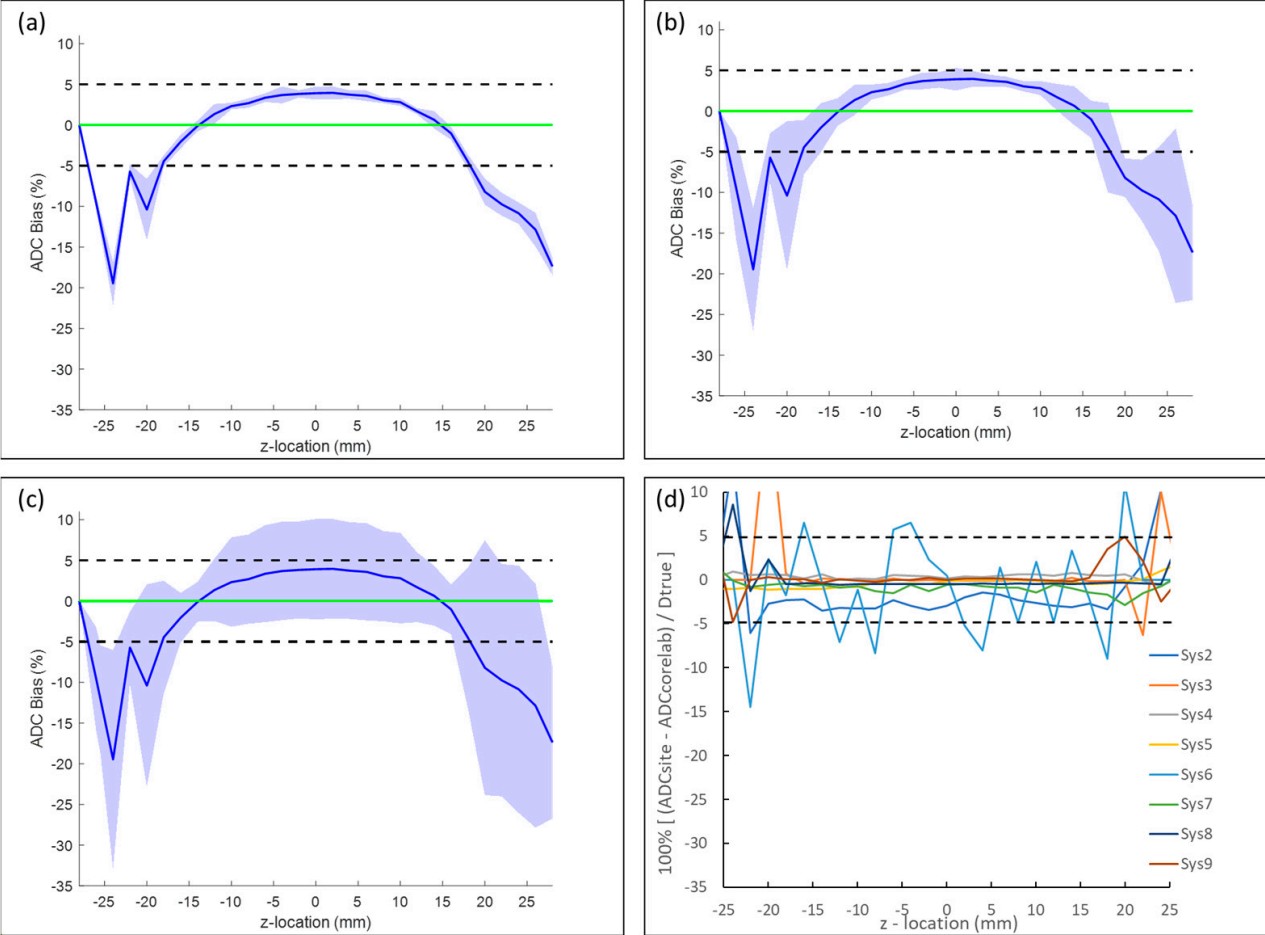

**Figure 4.** Summary of bias and repeatability for all studied systems: (**a**) mean bias (blue line) and short-term (intra-exam) repeatability relative to Dtrue, plotted as a function of *z*-axis location. Note, systems 8 and 9 did not provide short-term repeatability data (Table 1). (**b**) Corresponding plots for long-term (inter-exam) repeatability. Shaded regions in (**a**) and (**b**) represent bias $\pm 100\%$ *wSD*/Dtrue. (**c**) Cross-system reproducibility, where shaded region represents bias $\pm 100\%$ *SD*/Dtrue. Green line denotes ideal 0% bias. (**d**) Difference between site-generated and core-lab-generated ADC relative to Dtrue. Difference for core-lab systems 1 and 10 is zero (not plotted). Plots are on the same scale to aid visual comparison of bias, short- and long-term repeatability, reproducibility, and difference between site- versus core-lab ADC generation routines.

**Table 2.** Summary of isocenter ADC statistics across all systems with [95% confidence intervals].

| Short-Term Repeatability | | | Long-Term Repeatability | | | Cross-System Reproducibility | |
|---|---|---|---|---|---|---|---|
| *wSD* ($\mu m^2$/ms) | *RC* ($\mu m^2$/ms) | *wCV* (%) | *wSD* ($\mu m^2$/ms) | *RC* ($\mu m^2$/ms) | *wCV* (%) | *SD* ($\mu m^2$/ms) | *RDC* ($\mu m^2$/ms) |
| 0.009 [0.007, 0.014] | 0.025 [0.018, 0.038] | 0.73 [0.54, 1.12] | 0.015 [0.011, 0.023] | 0.042 [0.032, 0.064] | 1.26 [0.94, 1.89] | 0.068 [0.047, 0.124] | 0.188 [0.129, 0.343] |

All data in Figure 4a–c were derived from the core-lab-generated ADC maps. Figure 4d illustrates the percent difference between the core-lab-generated and the site-generated ADC values relative to Dtrue. Sys1 and Sys10 are not plotted in Figure 4d since these were core lab MRI systems, and thus would show "0%" difference. There were large random differences at peripheral slices (|z-offset| > 20 mm), potentially due to how various site algorithms deal with low SNR conditions. Of greater interest and significance were the clear ADC discrepancies within ≈15 mm of the isocenter, since measurements were derived from the very same good-quality DWI data. Root-mean-square differences within ±16 mm of the isocenter, between core-lab and site processing, were negligible (<0.6%) for five systems (Sys3, Sys4, Sys5, Sys8, and Sys9); ≈1% for Sys7; ≈3% for Sys2; and ≈5% for Sys6.

## 4. Discussion

Standardization of DWI acquisition and data processing protocols is essential to identify and mitigate technical sources of variance to enhance the scientific yield in preclinical and clinical studies. Even with consensus on primary acquisition parameters, platform and scanner-specific idiosyncrasies in protocol implementation can lead to unanticipated variance from system instability and chronic effects such as gradient nonlinearity and amplitude miscalibration. The pattern of peak ADC at/near the isocenter (z = 0) that falls off with |z| distance along the bore axis, as observed on these preclinical systems, is consistent with gradient nonlinearity observed on horizontal-bore clinical MRIs [26,41,42] due to known physical characteristics of gradient coils. In the context of quantitative ADC, gradient nonlinearity results in a spatially variable *b*-value. Secondary acquisition factors (e.g., shim routine and subject positioning) add yet more variance.

In this study, the CIRP IADP WG sought to determine base-level technical variance in performing ADC measurements on preclinical MRIs. To achieve this, a shared temperature-controlled DWI phantom with known diffusivity was used along with a detailed phantom preparation and scan procedure. To reduce variance due to data processing, centralized analysis was used to assess bias, short-term and long-term repeatability, and cross-system reproducibility. A key finding of this work was that performance of preclinical MRIs at isocenter resembles clinical MRIs [22] in terms of low average bias at isocenter (<4%), good repeatability (short-term *wCV* = 0.73%; long-term *wCV* = 1.26%), and cross-system reproducibility (*SD* = 0.068 $\mu m^2$/ms or 6.2% of Dtrue). Spatial nonuniformity of ADC measurements along the *z*-axis on preclinical MRIs also resembles the gradient nonlinearity observed on human MRIs [26,42], though scaled for bore size. While reasonable ADC uniformity over the central region (within ≈10 mm of the isocenter) was observed for most systems, the importance of repeatable subject (mouse) positioning of the organ/lesion of interest at/near isocenter must not be overlooked.

In terms of bias at the isocenter, it is clear that systems 9 and 10 are the dominant contributors to bias in this study. The aggregate CIRP system bias of <4% reported here would be reduced to <1.5%, and cross-system variability improved from *RDC* = 17% to 3% if these two outlier systems were excluded from analysis. Systems 9 and 10 happened to be at field-strength extremes, though we expect this is incidental and not the source of their bias. Elevated phantom temperature and data processing were eliminated as sources of bias since ample ice surrounded the measurement tubes and there was excellent agreement between site- and core-lab-generated ADC results for system 9. Of the CIRP scanners evaluated, system 9 operated on the oldest Bruker software version and system 10 was

the only MR Solutions platform, which may be contributing factors along with gradient amplitude miscalibration. Vendor-provided directional *b*-values were used for core-lab ADC map generation, although only system 10 (MR Solutions) *b*-values were numerically identical to nominal *b*-values, suggesting that calibrated values are perhaps unknown for this system.

Multiple studies of bias/repeatability/reproducibility of ADC on clinical MRIs using DWI phantoms are available [22–26]. Spatial nonuniformity of ADC as a function of x–y offset, as well as z-location on clinical scanners, has been studied using specialized phantoms and procedures [43–46] within FOVs much larger than typical for preclinical scanners. In our study, we limited ADC nonuniformity measurements to small offsets ($\pm 25$ mm) relevant for mouse DWI along the bore axis (z-direction), since this typically is the most nonuniform direction on clinical scanners, as predicted by horizontal bore gradient coil design specifications [42–45]. The only other prior work on multisystem preclinical MRIs [27] did not specifically address spatial nonuniformity. In addition, while the repeatability and reproducibility results of our study were comparable to the prior work [27], the overall system bias was much lower in our study. Reduced overall bias may be the result of improved system calibration procedures, updated scanner software, and/or use of a standardized acquisition protocol with centralized data processing. Our study used higher maximum *b*-value than the previous work [27], and inadequate SNR at high *b*-values is known to lead to ADC underestimation. However, our analysis showed that the average ADC bias at isocenter was positive with respect to true diffusion value, suggesting that low SNR was not a major contributor to the measured bias. Furthermore, our simulations for the observed SNR > 150 (at low *b*-value) indicated that our ADC results could not be significantly biased by Rician noise. Lastly, ADC was overestimated for systems 9 and 10, which dominated bias (as opposed to underestimating ADC, as predicted by low-SNR simulation). This suggested gradient miscalibration as a more likely source of the detected bias, possibly similar to systems in the prior multi-scanner study [27].

Observed discrepancies between a few site-generated and the core-lab-generated ADC values were greater than those explainable by noise or slight shift in ROI location. Fit algorithm details (e.g., log-linear versus nonlinear fit with possible accommodation of noise), use of nominal versus scanner-specific calibrated *b*-values, and/or incorrect scaling of scanner output are potential contributors to the detected deviations. In this study, core-lab processing utilized directional *b*-values discovered within Bruker and Agilent native data formats, although only nominal *b*-values were available in MR Solutions output data files. Except for the system 4's enhanced DICOM, which contained the *b*-matrix, all other ITK DWI datasets did not contain diffusion b-values and direction information; thus, one would need to know and use nominal *b*-values for quantitative ADC map generation. Site processing of system 6 was the most disparate relative to core-lab processing, particularly in terms of high slice-by-slice variability in ADC. Inspection of the system 6 ADC maps provided in classic DICOM revealed that the rescale slope (DICOM tag (0028,1053)) varied substantially (by $\approx$50%) with the slice number. Ignoring the rescale slope was the likely source of the high slice-by-slice variability observed in the system 6 site-based measurements [47]. These observations underline the importance of a standardized ADC generation workflow, along with standardized acquisition protocols and metadata (*b*-value and scale) recording.

## 5. Conclusions

Well-calibrated preclinical MRI systems are capable of highly repeatable and reproducible ADC measurements with low bias using standardized DWI data acquisition and processing protocols. Base technical-level repeatability and reproducibility metrics and spatial uniformity patterns are comparable to those observed on human systems using similar phantoms and test procedures.

**Supplementary Materials:** The following supporting information can be downloaded at: https://www.mdpi.com/article/10.3390/tomography9010030/s1, Figure S1: Measured SNR with simulated random noise and bias ADC error; Figure S2: Bland-Altman analysis excluding outlier scanners (systems 9 and 10) for short- and long-term ADC repeatability.

**Author Contributions:** Conceptualization, D.M. and T.L.C.; methodology, all authors; software, D.M., T.L.C., S.P., J.D.Q., P.L., R.G.P., G.A. and M.M.; validation, D.M. and T.L.C.; formal analysis, D.M. and T.L.C.; investigation, D.M., T.L.C., S.P., J.D.Q., P.L., R.G.P., S.T.G., G.A. and M.M.; data curation, D.M., T.L.C., S.P., J.D.Q., P.L., R.G.P. and M.M.; writing—original draft preparation, D.M. and T.L.C.; writing—review and editing, all authors; funding acquisition, T.L.C., R.Z., H.C.M., K.I.S., R.S., M.T.L. and P.E.K. All authors have read and agreed to the published version of the manuscript.

**Funding:** This research was funded by National Institutes of Health (grant numbers: U01CA166104, U24CA237683, U24CA231858, U24CA220325, U24CA209837, U24CA253531, S10OD026912, U24CA209837, U24CA253377, U24CA226110, U24CA264044, R50CA211270, and R01CA190299).

**Institutional Review Board Statement:** Not applicable.

**Informed Consent Statement:** Not applicable.

**Data Availability Statement:** The MRI datasets generated and analyzed for the current the study are available in MHD format from the corresponding author (T.L.C.) upon reasonable request.

**Conflicts of Interest:** T.L.C. and D.M. are coinventors of patents assigned to and managed by the University of Michigan.

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
