# Peer review of "Evaluation of Apparent Diffusion Coefficient Repeatability and Reproducibility for Preclinical MRIs Using Standardized Procedures and a Diffusion-Weighted Imaging Phantom"

_tomography, doi:10.3390/tomography9010030_

Round 1
Reviewer 1 Report
This work presents a study on the inter-system reproducibility of ADC measurements on pre-clinical MR scanners. The potential significance of the study is evident, as ADC values are used to quantify water mobility, one of the (few) quantitative biomarkers obtainable from MRI.
The paper is well written in general, and no objective errors were found. In my opinion, the main drawback of the presented work is that the study was performed only on preclinical systems (with small bore diameter), but conclusions are written for both, preclinical and clinical systems. While there are some references to literature on this point, I think the remaining difficulties for translating the findings to clinical systems are not properly discussed. For example, clinical systems allow for a much larger FOV in xy-direction, and, therefore the data will suffer more from imperfections in x-y-gradients, which cannot be studied with the small tube phantom used here.
Further, I have the following minor comments and suggestions:
Do not use (undefined) abbreviations in title and abstract (ADC and DWI; MRI is ok, I guess).
English writing: many articles missing, space between number and unit
L91: OD not defined, I assume outer diameter
L95: Give temperature of “conventional” freezer
L133 use shared cloud storage (or similar) instead of DropBox
L192: Move first three sentences of Results and Table 1 to Methods. In Table 1, add information about the used RF coils.
L296: center vs central
L299: “that” missing
L324: period missing
Fig. 4, caption: State that it shows the combined data from all 10 systems. In d) Add the explanation why Sys 1 and 10 are not plotted also here.
Author Response
We were pleased to receive constructive reviews of our manuscript. Below we list each Reviewer’s criticisms in italics along with point-by-point responses.
R1C1: “In my opinion, the main drawback of the presented work is that the study was performed only on preclinical systems (with small bore diameter), but conclusions are written for both, preclinical and clinical systems. While there are some references to literature on this point, I think the remaining difficulties for translating the findings to clinical systems are not properly discussed. For example, clinical systems allow for a much larger FOV in xy-direction, and, therefore the data will suffer more from imperfections in x-y-gradients, which cannot be studied with the small tube phantom used here.”
Response: The Reviewer correctly notes that this article is focused on characterizing pre-clinical scanners, though in the Discussion we comment on relevance of preclinical results to clinical systems, which is consistent with the co-clinical trial theme of this Tomography Special Issue. We cite multiple prior studies on DWI/ADC performance of clinical scanners that used analogous procedures with ice water-based DWI phantoms in original references 20, 22, 23, 24, and 26. We believe these provide comprehensive characterization of clinical MRIs, and a quality control standard for evaluation of multiple systems in the domain of DWI use in clinical trials (e.g., original refs. 9, 11). Rather than add to clinical MRI literature on this topic, our work addresses the paucity of studies on ADC reproducibility and bias of preclinical systems, currently represented by only one prior article (original ref. 27). Moreover, this prior work indicates a relatively large positive ADC bias on pre-clinical MRI systems, which our study refutes. We modified the Discussion to more explicitly point to the gap between study of clinical and pre-clinical systems. In the revised Discussion, we further elaborate on spatial non-uniformity of ADC in x-y and z-directions already well documented on clinical MRI systems (new references 43-46). Our pre-clinical study focused on non-uniformity along the bore axis (z-direction), since this typically is the most non-uniform direction for the horizontal bore gradient coil designs.
R1C2: “Do not use (undefined) abbreviations in title and abstract (ADC and DWI)”
Response: Abbreviations were expanded in Title and Abstract first mention, as requested.
R1C3: “many articles missing, space between number and space between number and unit”
Response: We reviewed the text for missing articles and removed spaces between numbers and units.
R1C4 (combined line edits): “L91: OD not defined, I assume outer diameter; L95: Give temperature of “conventional” freezer; L133 use shared cloud storage (or similar) instead of DropBox; L192: Move first three sentences of Results and Table 1 to Methods; In Table 1, add information about the used RF coils; L296: center vs central; L299: “that” missing ; L324: period miss”
Response: L91, L299, L324 edits were implemented as suggested. L95: added conventional freezer temperature of -18oC. L133: clarified that DropBox storage was used during the project per institution policy. L192: the first 3 sentences in Results and Table 1 were moved to Methods. Information on RF coils was not recorded and not added since RF coils do not directly influence diffusion weighting. L296: for clarification, wording was changed from “central »20mm” to “central region (within »10mm of isocenter)”.
R1C5: Fig. 4, caption: State that it shows the combined data from all 10systems. In d) Add the explanation why Sys 1 and 10 are not plotted also here
Response: Figure 4 caption was modified to “Summary of bias and repeatability for all studied systems. Difference for core lab Systems 1 and 10 is zero (not plotted).”
Reviewer 2 Report
The manuscript assessed the reproducibility and repeatability of ADC measurement on CIRP program pre-clinical MRI systems. Overall, it was well-written, and the methods were sufficiently described. A good agreement of ADC values was observed across different systems. Several parameters clearly support this conclusion: isocenter ADC bias, short-term/long-term repeatability, and cross-system reproducibility.
I have three concerns about the manuscript: 1) There are similar works in previous studies. Can the authors highlight the advances or new findings of this manuscript? 2) Systems 9 and 10 were found to be outliers. In order to show the ADC bias is based on the system, is that possible to provide results from more MRI systems similar to Systems 9 and 10? Also, should these systems be avoided in the program involving the other 8 systems? 3) I have noticed Scan2 of System 8 was missing. Does it mean that System 8 was not included in the short-term repeatability calculation?
Author Response
We were pleased to receive constructive reviews of our manuscript. Below we list each Reviewer’s criticisms in italics along with point-by-point responses.
R2C1: “There are similar works in previous studies. Can the authors highlight the advances or new findings of this manuscript?.”
Response: There is only one multi-system study reference for preclinical systems (original ref. 27). In Discussion we emphasize that our study evaluates spatial bias patterns and reports on much lower biases on most contemporary preclinical systems compared to previous results, likely reflecting improved gradient calibration.
R2C2: “Systems 9 and 10 were found to be outliers. In order to show the ADC bias is based on the system, is that possible to provide results from more MRI systems similar to Systems 9 and 10? Also, should these systems be avoided in the program involving the other 8 systems?”
Response: Figure 3 shows individual system biases, and Figure 4 shows the average bias across all systems (including the outliers). Although our study did not provide other examples of systems having same hardware/software properties as the two outlier scanners (systems 9 and 10), it is possible that systems used in previous study (original ref. 27) had similar deficiencies. It is also possible that observed biases will be similar for systems using the same gradient models and versions of gradient calibration software (listed in Table 1). Since observed biases for outliers were large at isocenter, they were most likely related to gradient miscalibration. For multi-site programs such systems could be disqualified or their data would need to be corrected using the ice-water phantom based gradient re-calibration.
R2C3: “I have noticed Scan2 of System 8 was missing. Does it mean that System8 was not included in the short-term repeatability calculation?”
Response: Table 1 shows that system 8 and 9 did not provide the second scan on both scan days, and hence were not used for short-term repeatability calculations as now clarified in Figure 4 caption.